# Specifics of Young Gastric Cancer Patients: A Population-Based Analysis of 46,110 Patients with Gastric Cancer from the German Clinical Cancer Registry Group

**DOI:** 10.3390/cancers14235927

**Published:** 2022-11-30

**Authors:** Markus Kist, Michael Thomaschewski, Yannick Keck, Thaer S. A. Abdalla, Sylke Ruth Zeissig, Kees Kleihues-van Tol, Ulrich Friedrich Wellner, Tobias Keck, Jens Hoeppner, Richard Hummel

**Affiliations:** 1Department of Surgery, University Medical Center Schleswig-Holstein, Campus Lübeck, Ratzeburger Allee 160, 23538 Lübeck, Germany; 2German Cancer Registry Group of the Society of German Tumor Centers—Network for Care, Quality and Research in Oncology (ADT), 14057 Berlin, Germany; 3Institute of Clinical Epidemiology 12wand Biometry (ICE-B), University of Würzburg, 97080 Würzburg, Germany

**Keywords:** gastric cancer in young patients, german clinical cancer registry group, early-onset gastric cancer patients

## Abstract

**Simple Summary:**

The incidence of gastric cancer shows marked age variations, and it is most frequently diagnosed in middle-aged and elderly patients between the ages 50 and 70. However, 2–8% of all gastric cancer occurs at a younger age, also known as early-onset gastric cancer. Studies characterizing this cohort of young patients with gastric cancer are scarce. The prognosis for this group of young patients with gastric cancer remains controversial, and it is unclear how to define the “young patient”. The objective of this study was to characterize age variations of gastric cancer in the German population using population-based registry data from the German Cancer Registry Group of the Society of German Tumor Centers—Network for Care, Quality and Research in Oncology (ADT)and to investigate whether a cohort of young patients can be identified who differ from elderly patients in terms of tumor stage at diagnosis, histology, and prognosis. In this study, we were able to objectively identify a cohort of patients referred to as early-onset gastric cancer by applying an approach that stratified relative distributions of histological subtypes of gastric adenocarcinoma according to age percentiles. With a median age of 53 years, this group of young patients showed more aggressive and advanced tumors and received significantly less curative treatment. However, survival of this early-onset gastric cancer patients was significantly better compared to elderly patients, both in general as well as stratified according to treatment. Young age was identified as an independent predictor for better survival in this study.

**Abstract:**

Introduction: 2–8% of all gastric cancer occurs at a younger age, also known as early-onset gastric cancer (EOGC). The aim of the present work was to use clinical registry data to classify and characterize the young cohort of patients with gastric cancer more precisely. Methods: German Cancer Registry Group of the Society of German Tumor Centers—Network for Care, Quality and Research in Oncology (ADT)was queried for patients with gastric cancer from 2000–2016. An approach that stratified relative distributions of histological subtypes of gastric adenocarcinoma according to age percentiles was used to define and characterize EOGC. Demographics, tumor characteristics, treatment and survival were analyzed. Results: A total of 46,110 patients were included. Comparison of different groups of age with incidences of histological subtypes showed that incidence of signet ring cell carcinoma (SRCC) increased with decreasing age and exceeded pooled incidences of diffuse and intestinal type tumors in the youngest 20% of patients. We selected this group with median age of 53 as EOGC. The proportion of female patients was lower in EOGC than that of elderly patients (43% versus 45%; *p* < 0.001). EOGC presented more advanced and undifferentiated tumors with G3/4 stages in 77% versus 62%, T3/4 stages in 51% versus 48%, nodal positive tumors in 57% versus 53% and metastasis in 35% versus 30% (*p* < 0.001) and received less curative treatment (42% versus 52%; *p* < 0.001). Survival of EOGC was significantly better (five-years survival: 44% versus 31% (*p* < 0.0001), with age as independent predictor of better survival (HR 0.61; *p* < 0.0001). Conclusion: With this population-based registry study we were able to objectively define a cohort of patients referred to as EOGC. Despite more aggressive/advanced tumors and less curative treatment, survival was significantly better compared to elderly patients, and age was identified as an independent predictor for better survival.

## 1. Introduction

Gastric cancer (GC) is one of the most common malignancies worldwide, with an incidence of about one million cases and 769,000 cancer-related deaths in 2020. Hence, GC ranks fifth for yearly incidence and fourth for yearly cancer-related mortality [1]. GC is most frequently diagnosed in middle-aged and elderly patients between the ages of 50 and 70, and in western countries more than half of all patients with GC are older than 70 years old [2,3,4,5]. However, 2–8% of GC occurs at a younger age, also known as early-onset gastric cancer (EOGC) [6,7]. Within recent years, the incidence of GC in middle-aged and elderly patients has steadily decreased, based for example on eradication of H. pylori or improved preservation and storage of food [1,8]. In contrast, stable or even slightly increasing incidences have been reported for EOGC [9].

Studies on EOGC are limited in numbers and by the fact that definitions of EOGC vary widely across the literature [2,10,11,12]. So far, more aggressive tumor behavior with advanced tumor stages at time of diagnosis, undifferentiated histology and lymphovascular invasion has been reported [13]. However, prognosis of these young patients remains controversial, as some studies reported poorer overall survival compared to middle-aged and elderly patients [11,14,15,16], whereas more recent studies found equivalent and even improved survival of this cohort [10,17,18].

Based on the lack of high-level evidence, the current study now aimed to define and characterize EOGC in a population-based analysis of data from the German Cancer Registry Group of the Society of German Tumor Centers—Network for Care, Quality and Research in Oncology (ADT).

## 2. Methods

### 2.1. Study Population

This retrospective population-based study is based on the German Cancer Registry Group of the Society of German Tumor Centers—Network for Care, Quality and Research in Oncology (“Arbeitsgemeinschaft Deutscher Tumorzentren e.V.” (ADT)), currently with 61 members. Together with the German Cancer Society (DKG) and the German Cancer Aid (DKH), the ADT aims to improve cancer care in Germany. In this context, the German Clinical Cancer Registry Group has been implemented with the purpose to collect and combine a predefined data set for various cancer entities [19], from regional clinical cancer registries all over the country for health services research and quality assurance on a voluntary basis.

For the purpose of this study, we queried the German Clinical Cancer Registry Group for data on gastric cancer patients (ICD-10: C16) between 2000 and 2016. Given several clinically and oncologically relevant differences between tumors at the cardia/gastroesophageal junction and tumors at other sites in the stomach (such as differing histology/histological subtypes, differences in gender and age distribution or differences in treatment algorithms, amongst others), we decided to exclude specifically tumors at the cardia/gastroesophageal junction (ICD 16.0) from our analysis and to focus on gastric cancers in the remaining stomach (ICD-10: C16.1–C16.9).

### 2.2. Study Parameters

Data were assessed for plausibility in order to exclude duplicates, patients with missing data on date of diagnosis or birth or sex, patients with death certificates only, date of death before date of diagnosis, date of diagnosis not within 2000–2016 and ICD-10 codes other than C16.1–C16.9. Study parameters included demographics (age calculated from date of diagnosis and the date of birth), sex, tumor classification (histology according to ICD-O-3, differentiation, clinical pretherapeutic TNM classification (modified), ICD-10 code), treatment modalities (curative intended treatment options included: neoadjuvant treatment + curative surgery, curative surgery + adjuvant treatment, curative surgery + chemo- and/or radio-therapy, curative surgery without further documented treatment, and definitive radio-chemotherapy; palliative intended treatment options included: palliative surgery + chemo- and/or radio-therapy, palliative surgery without further documented treatment, chemo- and/or radio-therapy without intention provided), and patient follow-up (90-day mortality, calculated on the basis of date of surgery and date of death, only calculated for patients undergoing curative surgery; overall survival (OS), calculated from date of diagnosis and date of last follow up or date of death). Details on study parameters are provided in Appendix A.

### 2.3. Definition of Early-Onset Gastric Cancer Patients

In order to identify early-onset gastric cancer patients, we first searched for an early peak of disease incidence by analyzing distribution of age in the entire patient population as well as in the three major histological adenocarcinoma subtypes (intestinal type/diffuse type/signet ring cell gastric carcinoma (SRCC)). Next, we analyzed the relative distribution of the three major histological adenocarcinoma subtypes within the population. Finally, we stratified these relative distributions according to age percentiles and selected a group of patients in whom the relative incidence of SRCC exceeded that of intestinal and diffuse type carcinomas together. This approach using percentiles rather than pre-defined age cut-off points (e.g., <50 years of age, <60 years of age) was applied in order to avoid “random selection” of an age cut-off point.

### 2.4. Statistics and Ethics

Data processing and statistical analysis were performed using R (version 3.5.1). Ordinal and nominal variables were expressed as absolute numbers and percentages. The Chi-square-test was used for comparison of ordinal and nominal variables between groups. Numeric variables were expressed as means with a confidence interval (CI) or as median with an interquartile range (IQR). Pie and bar charts were used to visualize results. Overall survival was analyzed using the Kaplan-Meier method, and the Log-rank test was used for statistical comparison. Uni- and multi-variate Cox regression analysis were used to determine independently predictive variables for better or worse survival. For all statistical analyses, a *p*-value of *p* ≤ 0.050 was considered significant.

The study was reviewed and approved by the Institutional Ethics Committee (Ethik-Kommission Universität zu Lübeck/Aktenzeichen: 20-237). Further, the study was approved by the ADT.

## 3. Results

### 3.1. Study Population

A total of 46,110 patients with a malignant neoplasm of the stomach (C16.1–C16.9) were documented in the ADT German Clinical Cancer Registry Group between 2000 and 2016. Based on information of the Association of Population Based Cancer Registries in Germany (GEKID), this accounts for approximately 20% of all gastric cancer cases in this period in Germany.

Demographics of the entire study population showed that 45% of the patients were female and 55% were male (ratio 1.22:1 male to female). The median age of patients at diagnosis was 72 years, with women being slightly older than men (74 versus 71 years).

### 3.2. Definition of Patients with Early-Onset Gastric Cancer

Distribution of age in the entire population as well as distribution of age in the three major histological adenocarcinoma subtypes (intestinal type, diffuse type, and signet ring cell gastric carcinoma (SRCC)) is shown in Figure 1. The entire population presented a peak at 74 years with a median age of 72 years (IQR 63–80 years). All three major histological adenocarcinoma subtypes presented similar peaks at about 74 years. Neither the entire patient population nor any of the major histological adenocarcinoma subtypes presented apart from the major peak at an advanced age a second distinct peak in early years. However, the median age of patients differed significantly between histological subtypes, with SRCC patients presenting the youngest median age (67.2 years; IQR 56–76), followed by patients with diffuse type carcinoma (70 years; IQR 60–78) and patients with intestinal type carcinoma (75 years; IQR 68–82).

Based on the difference in median age between the three major histological adenocarcinoma subtypes, we next analyzed the relative distribution of the three major histological adenocarcinoma subtypes within the population stratified according to age percentiles. And in fact, we found with decreasing age a significant increase of the portion of SRCC patients in the population (Figure 2). SRCC was diagnosed in 41% of the youngest 50% of patients. The portion of SRCC further increased to 49% (youngest 20%), 53% (youngest 10%), and went up to 57% in the youngest 5% of patients.

From an age below approximately 60 years, the portion of SRCC in the patient population exceeded 50%, hence this histological subtype was more frequent than the other two subtypes put together. As this “cut off” was close to the cut off between the youngest 20% and oldest 80% of patients, we selected the youngest 20% of patients as the “early-onset GC” (EOGC) in this study and the remaining oldest 80% of patients as “late-onset GC” (LOGC).

### 3.3. Characterization of Early-Onset GC Patients

9221 patients were allocated into the EOGC group, and 36,889 patients into the LOGC group. The median age of the EOGC group was, at 53 years, >20 years lower than that of LOGC (75 years). The proportion of female patients was significantly lower in EOGC than in LOGC with 43% versus 45% (*p* < 0.001). Adenocarcinoma was the major tumor type in both groups, but was significantly more frequent in LOGC than in EOGC (90 versus 87%, *p* < 0.001). Gastrointestinal stromal tumors (GIST) and neuroendocrine tumors were significantly more often diagnosed in EOGC than LOGC (3.4% versus 2.6%, and 3.9% versus 2.1%, respectively; *p* < 0.001).

Table 1 presents an overview about tumor characteristics in both groups. In summary, EOGC patients presented significantly more advanced and undifferentiated tumors compared to LOGC patients, with G3/4 stages in 77% versus 62%, T3/4 stages in 51% versus 48%, nodal positive tumors in 57% versus 53%, and metastasis in 35% versus 30%.

Figure 3 shows treatment algorithms in EOGC and LOGC. EOGC patients were receiving curative treatment significantly less frequently (curative treatment in general: 42% versus 52%; curative surgery with/without further treatment: 36% versus 49%; *p* < 0.001). A closer look at specific treatments in curative settings especially reveals that the rate of curative surgery without further documented treatment in EOGC was nearly half of that in LOGC. Most other curative treatment options (especially multimodal treatment with either neoadjuvant or adjuvant treatment, or definitive radio-chemotherapy) have in fact been applied more frequently in EOGC.

The median overall survival of the entire study population was 21.9 months (95% CI: 21.3/22.3) with a 5-year survival rate of 33.5% (95% CI: 32.9/34.1). Interestingly, median survival was significantly better in EOGC patients with 38.6 months (95% CI 34.6/42) compared to 19.2 months (95% CI: 18.3/19.7; *p* < 0.0001) in LOGC. Accordingly, EOGC presented significantly higher 5-years survival rates of 44.0% (95% CI: 42.7/45.3) compared to 30.6% (95% CI: 30.0/31.3) in LOGC (*p* < 0.0001, Figure 4).

Median overall survival after curative surgery with/without further treatment was 76.4 months (95% CI: 70.6/82.6), with 5-year survival of 56.0% (95% CI: 54.2/57.7) for the entire population after exclusion of early 90-day mortality, and median survival after palliative treatment was 15.0 months (95% CI: 14.2/15.3), with five-year survival of 16.3% (95% CI: 15.0/17.6). Again, EOGC patients presented significantly better survival after exclusion of early 90-day mortality after both curative surgical and palliative treatment, but the differences were less distinct after palliative treatment (Table 2). Notably, 90-day mortality was significantly higher in LOGC patients (6.7% versus 1.9%; *p* < 0.001).

Univariate Cox regression analyses on the entire population revealed a significant impact of age, histological tumor type, histological subtypes in adenocarcinoma (SRCC versus diffuse type versus intestinal type), differentiation and TNM stage on overall survival [20]. Results of multivariate Cox regression analyses on the entire patient population, including only patients with adenocarcinoma histology, are shown in Table 3. In this context we were able to demonstrate that age, histological subtypes in adenocarcinoma (SRCC versus diffuse type versus intestinal type), differentiation, and TNM stage were independent predictors of survival, but sex was not. Interestingly, in patients with curative intended surgery with/without further treatment, histological subtype and differentiation was not confirmed to independently predict survival, and histological subtype and node positive disease was no independent predictor of survival in palliative settings. However, young age (EOGC) was in both curative and palliative settings an independent predictor of better survival.

## 4. Discussion

In Western countries, gastric cancer is most frequently diagnosed in elderly patients [2,3,4,5]. However, there is a clinically well-known and often-described cohort of patients who develop gastric cancer at a young age, also referred to as early-onset gastric cancer (EOGC). To date, the characteristics of this patient group are poorly studied, as occurrence of gastric cancer at a young age is rare, and definitions of EOGC vary widely across the literature [11,14,15,21]. In this present work, we used data from the German Cancer Registry Group of the Society of German Tumor Centers—Network for Care, Quality and Research in Oncology (ADT)to define and characterize EOGC in a population-based analysis. A total of 46,110 patients diagnosed with gastric cancer (C19.1–C16.9) between 2006 and 2016 were included into this study, corresponding to approximately one-fifth of all gastric cancer patients in Germany in this period.

In the literature, definition of EOGC is usually characterized only by (random) selection of age or of a portion of the entire population such as “the youngest 5% of patients”. This leads to enormous heterogeneity in the classification of EOGC, which is defined, for example, as all patients < 34 years of age, or patients < 40 years or <45 years) [11,14,15,21]. To our best knowledge, only one study used additional characteristics of patients (survival) to define this population [22]. Given the fact that there is, apart from the major peak of incidence of GC at 74 years, no further peak in early years, definition of EOGC based on age alone seems unreasonable. Therefore, we applied in this study an approach that stratified relative distributions of histological subtypes of gastric adenocarcinoma according to age percentiles. In more detail, we compared different portions of the entire population based on age (youngest 5%, youngest 10%, youngest 20%, etc.) with incidence of the three major histological subtypes of gastric adenocarcinoma. With this approach, we could show that especially SRCC incidences increased significantly with younger age and exceeded the pooled incidences of diffuse and intestinal type tumors together at around 60 years of age. As this cut-off reflected the youngest 20% of patients, we selected this definition of EOGC for the current study. And, in fact, the median age of EOGC was, at 53 years, only slightly higher than that of most reports [11,14,15,21].

Characterization of EOGC then showed that proportion of female patients was significantly lower in EOGC than in late-onset GC (LOGC) with 43% versus 45%. Adenocarcinoma was the major tumor type in both groups, but its incidence differed significantly between EOGC and LOGC (87% versus 90%). Further, we found GIST and neuroendocrine tumors to be significantly more frequent in EOGC. With regards to the proportion of female patients in EOGC, our results seem to contradict most studies that showed more female patients in younger groups (14, 15, 21). As most studies investigated only adenocarcinoma patients, we hypothesized that inclusion of different histologies (adenocarcinoma, GIST, etc.) in our analysis might impact the distribution of female and male patients. However, using only data on adenocarcinoma patients, our results remained identical, with less female patients in EOGC (43% versus 45%). Most interestingly, we then found more detailed reports on proportions of female patients in different age groups from the SEER-registry. Al-Refaie and colleagues demonstrated proportion of female patients to be similar below 45 years and above 70 years (43% and 42%, respectively), but to be lower in patients between 45 and 70 years (32%) [10]. This basically reflects the results of the aforementioned studies when comparing combined patient groups of elderly patients (45–70 years plus > 70 years) to younger patients. And in fact, our data showed comparable results when comparing the same age groups (<45 years: 52%, 45–70 years: 39%; >70 years: 48%). Hence, our data are in accordance with reported results on proportion of female patients but differ slightly due to definition of EOGC. With regards to increased rates of GIST and neuroendocrine tumors in our EOGC population, we found no reports in the literature that included all tumor histologies in the stomach when assessing EOGC patients. This is, therefore, to our best knowledge, the first report to thoroughly assess all patients diagnosed with gastric cancer (C16.1–C16.9) with regards to different histologies in EOGC other than adenocarcinoma. Looking at adenocarcinoma subtypes, our study results are in accordance with other studies showing that SRCC incidences increased with decreasing age, and that the median age of SRCC patients is generally lower compared to other subtypes [23,24,25].

With regards to biological behavior, EOGC patients presented in the current study with significantly more advanced and undifferentiated tumors compared to LOGC patients. Again, this is consistent with multiple other reports on differentiation [10,26] or TNM stages [11,15]. Several reasons might explain these more aggressive and advanced tumors in EOGC. These include delayed diagnosis due to low GC incidence in younger patients [15,26], or higher rates of biologically more aggressive tumor subtypes such as SRCC in young patients [11,21].

The present study further demonstrated less frequent curative treatment in EOGC than in LOGC. Especially the rate of curative surgery without further documented treatment in EOGC was nearly half of that in LOGC, while most other curative treatment options (multimodal treatment or definitive radio-chemotherapy) have been applied more frequently in EOGC. These findings are somewhat reasonable as EOGC presented more advanced and aggressive tumors, and the results are partly supported by the literature. Data from SEER-registry showed patients <45 years of age as receiving significantly less frequently curative surgery compared to patients between 45 and 70 years (56% versus 58%). However, in the same study, patients > 70 years received significantly less curative surgical treatment compared to young patients (50%) [10].

Finally, our study showed that survival of EOGC is significantly better compared to LOGC, both in the entire population as well as in curative/palliative treated EOGC patients. The survival advantage of EOGC was more obvious in curative-treated patients compared to palliative treatment with differences in five-year survival rates of approximately 10% versus 5%. Furthermore, multivariate analyses on adenocarcinoma patients only demonstrated that age, histological subtype, differentiation, and TNM stage were independent predictors of survival, but sex was not. Interestingly, while histological subtype and differentiation respectively histological subtype and node positive disease were not confirmed to independently predict survival in curative surgery respectively palliative settings, young age (EOGC) was in both curative and palliative settings an independent predictor of better survival. While impact of for example histology, subtype, differentiation or TNM staging on survival is well established in further studies data on the relevance of age on prognosis remains controversial to this date [10,11,15,21,26,27]. Some studies reported poorer overall survival of EOGC compared to middle-aged and elderly patients [11,22], whereas more recent studies found equivalent and even improved survival of this cohort [10,21,24,27]. However, especially data from large SEER-registry studies confirmed our results with patients <45 years of age presenting better survival compared to patients >70/>66 years of age [10,27]. Furthermore, we identified age (EOGC versus LOGC) as an independent predictor for better survival, which was also confirmed, for example, for operable gastric cancer patients by Chen et al. [27]. Reasons for better survival of EOGC might include better performance status or better physical condition leading to better tolerance of (multimodal or surgical) treatment [27].

Although this study represents a large population-based registry study, a number of limitations have to be addressed for the proper interpretation of results. First, the German Clinical Cancer Registry Group originates from several regional clinical cancer registries, implicating data entry by many different people. This leads to partly incomplete or even incorrect data, especially with regards to treatment information. In addition, disease-free survival is included in the dataset but completeness and plausibility are lacking; therefore we decided not to include this variable into the analysis despite its high clinical relevance. Furthermore, based on the underlined predefined data set, there is a lack of several important points of information such as comorbidities, H. Pylori status, exact surgical details (for example extent of lymphadenectomy), postoperative complications, or details of histological examination. Thus, analyses could not be adjusted for potential confounding factors. Finally, in the attempt to objectively investigate if there is a young group of patients that can be identified not only by age, we used an approach that stratified relative distributions of histological subtypes of gastric adenocarcinoma according to age percentiles. This approach using percentiles rather than pre-defined age cut-off points (e.g., <50 years of age, <60 years of age) was specifically applied in order to avoid “random selection” of an age cut-off point. However, this approach limits generalizability of the results and comparison to other publications using age-defined categories.

## 5. Conclusions

In conclusion, with this large-scale population-based registry study including 46,110 patients with gastric cancer from German Cancer Registry Group of the Society of German Tumor Centers—Network for Care, Quality and Research in Oncology (ADT), we were able to objectively identify and define a cohort of patients referred to as early-onset gastric cancer by applying an approach that stratified relative distributions of histological subtypes of gastric adenocarcinoma according to age percentiles. With a median age of 53 years, this group of young patients presented more aggressive and advanced tumors and received significantly less curative treatment. However, survival of this early-onset gastric cancer patients was significantly better compared to elder patients, both in general as well as stratified according to treatment. Indeed, young age was identified as independent predictor for better survival in this study. These results highlight on the one hand the clear need for increased awareness of this disease in young patients. Stringent diagnostic workup in case of unspecific symptoms for example with early endoscopy might allow earlier identification of gastric cancer in these patients potentially leading to earlier tumor stages at time of diagnosis. On the other hand, our data can impact clinical decision making as we found favorable outcomes in these young patients even despite advanced tumor stages implicating liberal indication for aggressive curative (multimodal) treatment in this cohort.

## Figures and Tables

**Figure 1 cancers-14-05927-f001:**
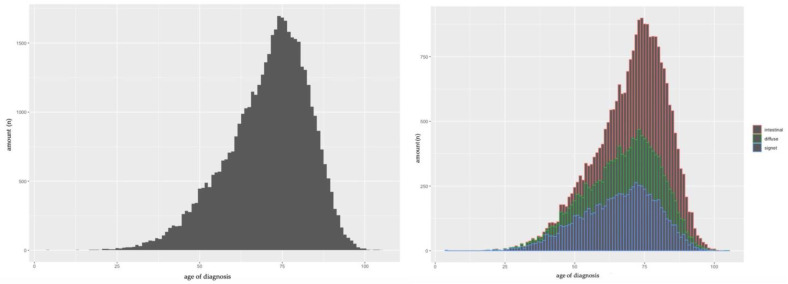
Gastric cancer population. Left: dispersion by age. Right: dispersion by age and histopathology; red = intestinal type, green = diffuse type, blue = signet type.

**Figure 2 cancers-14-05927-f002:**
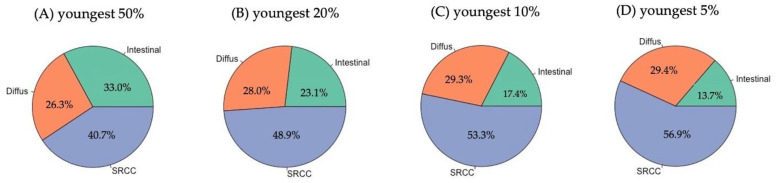
Relative distribution of the three major histological adenocarcinoma subtypes within the population stratified to age percentiles: (**A**) youngest 50% (<72.4 years); (**B**) youngest 20% (<60.4 years); (**C**) youngest 10% (<52.8 years); and (**D**) youngest 5% (<47.1 years).

**Figure 3 cancers-14-05927-f003:**
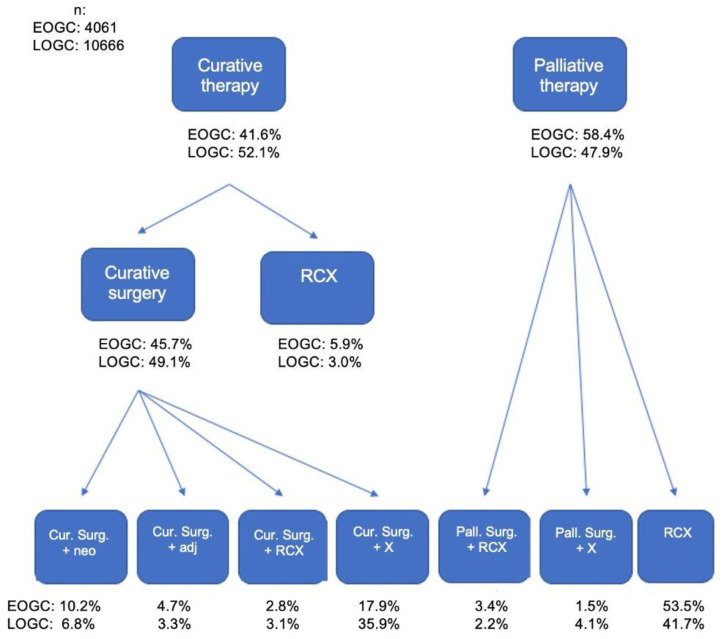
Treatment algorithms in both groups EOGC and LOGC.

**Figure 4 cancers-14-05927-f004:**
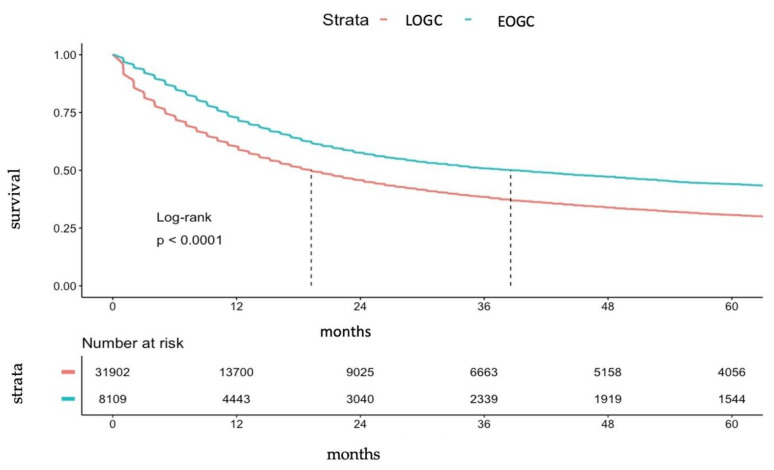
Kaplan-Meier-Curve for survival for EOGC and LOGC.

**Table 1 cancers-14-05927-t001:** Overview of tumor characteristics (TNM-/G-stage) in both groups.

TNM-Status	Stages	EOGC	LOGC	*p*-Value
Grading (*n* = 7573 versus 31,342)	G1	458	2162	*p* < 0.001
		6.0%	6.9%	
	G2	1292	9813	
		17.1%	31.3%	
	G3	5557	18,529	
		73.4%	59.1%	
	G4	266	838	
		3.5%	2.7%	
T stage (*n* = 5105 versus 18,835)	T1	1094	4344	
		21.4%	23.1%	
	T2	1425	5514	*p* < 0.001
		27.9%	29.3%	
	T3	1638	5817	
		32.1%	30.9%	
	T4	948	3160	
		18.6%	16.8%	
N stage (*n* = 5029 versus 18,641)	N0	2154	8860	*p* < 0.001
		42.8%	47.5%	
	N+	2875	9781	
		57.2%	52.5%	
M stage (*n* = 5140 versus 18,096)	M0	3356	12,751	*p* < 0.001
		65.3%	70.5%	
	M+	1784	5345	
		34.7%	29.5%	

**Table 2 cancers-14-05927-t002:** Median overall survival and five-year survival for curative surgery with/without further treatment and palliative treatment in EOGC and LOGC.

	EOGC		LOGC		
	Median Overall survival (months)	5-year survival	Median Overall survival (months)	5-year survival	*p*-value
Curative surgery with/without further treatment	128.7 (95% CI: 101.5/149.1)	63.6% (95% CI: 60.0/66.9)	68.2% (95% CI: 63.5/73.2)	53.5% (95% CI: 51.5/55.5)	*p* < 0.001
Palliative treatment	16.3 (95% CI: 15.2/17.3)	19.5% (95% CI: 17.2/21.9)	14.2 (95% CI: 13.2/14.9)	14.6% (95% CI: 13.1/16.2)	*p* < 0.001

**Table 3 cancers-14-05927-t003:** Multivariate Cox regression analyses on the entire patient population, including only patients with adenocarcinoma histology.

	Multivariat Regression Analysis *n* = 8976			Late Onset *n* = 6779			Early Onset *n* = 2197		
	Hazard ratio	95% CI	*p*-value	Hazard ratio	95% CI	*p*-value	Hazard ratio	95% CI	*p*-value
female	1			1			1		
male	1.02	0.96–1.09	0.444	1.01	0.95–1.08	0.755	1.09	0.96–1.24	0.174
LOGC/>60.4 years	1								
EOGC/<60.4 years	0.61	0.56–0.65	0.000						
Histological subtype									
intestinal type	1			1			1		
diffuse type	1.13	1.04–1.23	0.003	1.13	1.04–1.24	0.007	1.15	0.94–1.42	0.182
SRC type	1.10	1.02–1.20	0.018	1.09	1.00–1.20	0.058	1.15	0.95–1.41	0.169
Grading									
G1	1.03	0.84–1.27	0.778	1.03	0.83–1.28	0.785	0.80	0.39–1.63	0.530
G2	0.94	0.86–1.02	0.157	0.96	0.87–1.05	0.352	0.80	0.63–1.01	0.060
G3	1			1			1		
G4	1.33	1.15–1.55	0.000	1.27	1.06–1.52	0.009	1.44	1.10–1.88	0.007
T-status									
T1	0.49	0.44–0.55	0.000	0.55	0.49–0.62	0.000	0.31	0.24–0.41	0.000
T2	0.70	0.65–0.76	0.000	0.73	0.67–0.79	0.000	0.61	0.52–0.71	0.000
T3	1			1			1		
T4	1.28	1.18–1.39	0.000	1.30	1.18–1.43	0.000	1.19	1.00–1.41	0.049
N-status									
N0	1			1			1		
N+	1.53	1.42–1.65	0.000	1.58	1.45–1.71	0.000	1.38	1.17–1.62	0.000
M-status									
M0	1			1			1		
M1	2.38	2.23–2.57	0.000	2.24	2.06–2.43	0.000	2.84	2.45–3.28	0.000
	Multivariat regression analysis *n* = 8976			Curative-operative treatment (>90 days survival) *n* = 1467			Palliative treatment *n* = 1434		
	Hazard ratio	95% CI	*p*-value	Hazard Ratio	95% CI	*p*-value	Hazard Ratio	95% CI	*p*-value
demographics									
female	1			1			1		
male	1.02	0.96–1.09	0.444	1.02	0.87–1.20	0.792	1.03	0.91–1.18	0.638
o80/>60.4 years	1			1			1		
y20/<60.4 years	0.61	0.56–0.65	0.000	0.58	0.47–0.70	0.000	0.80	0.70–0.93	0.003
Histological subtype									
Intestinal type	1			1			1		
Diffuse type	1.13	1.04–1.23	0.003	1.11	0.89–1.40	0.347	1.15	0.96–1.38	0.117
SRC type	1.10	1.02–1.20	0.018	1.20	0.96–1.51	0.117	1.01	0.84–1.22	0.885
Grading									
G1	1.03	0.84–1.27	0.778	0.99	0.59–1.66	0.970	0.94	0.57–1.55	0.815
G2	0.94	0.86–1.02	0.157	0.84	0.66–1.06	0.136	0.84	0.69–1.03	0.088
G3	1			1			1		
G4	1.33	1.15–1.55	0.000	1.16	0.57–2.34	0.678	1.63	1.14–2.33	0.007
T-status									
T1	0.49	0.44–0.55	0.000	0.42	0.32–0.55	0.000	0.88	0.67–1.16	0.378
T2	0.70	0.65–0.76	0.000	0.69	0.57–0.83	0.000	0.81	0.68–0.96	0.015
T3	1			1			1		
T4	1.28	1.18–1.39	0.000	1.33	1.01–1.76	0.041	1.14	0.98–1.33	0.093
N-status									
N0	1			1			1		
N+	1.53	1.42–1.65	0.000	1.74	1.45–2.08	0.000	0.95	0.81–1.12	0.548
M-status									
M0	1			1			1		
M1	2.38	2.23–2.57	0.000	2.17	1.63–2.89	0.000	2.01	1.75–2.30	0.000

## Data Availability

Data were obtained from the German Cancer Registry Group of the Society of German Tumor Centers—Network for Care, Quality and Research in Oncology (ADT) and are available from the authors with the permission of The Society of German Tumor Centers.

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
