# Peer review of "Specifics of Young Gastric Cancer Patients: A Population-Based Analysis of 46,110 Patients with Gastric Cancer from the German Clinical Cancer Registry Group"

_cancers, 2022, doi:10.3390/cancers14235927_

Round 1

Reviewer 1 Report (Previous Reviewer 2)

The authors have addressed all the questions.

Reviewer 2 Report (Previous Reviewer 1)

I would like to thank the editor and the authors for giving me the opportunity to review this descriptive study on gastric cancer in Germany. The authors responded to all my comments, and they improved, in the manuscript, the transparency of the clustering method and extended the section with the limitation.

This manuscript is a resubmission of an earlier submission. The following is a list of the peer review reports and author responses from that submission.

Round 1

Reviewer 1 Report

I would like to thank the editor and the authors for giving me the opportunity to review this descriptive study on gastric cancer in Germany. The study evaluated age at diagnosis, tumor stage, histology, and prognosis of 46,110 patients with gastric cancer from the German Clinical Cancer Registry Group. The authors claimed to have defined a subgroup of early-onset gastric cancer based on age at diagnosis and histology. However, I found the manuscript not very transparent regarding the methodological method behind the clustering. In addition, conclusions can be directly function of the assumptions made in the analysis (comparing Kaplan-Meier curve between age-stratified patients groups). Please find here below some major and minor comments.

Major:

11. Methods. The clustering of the patients is not described in the methods.

22. Methods. Given the available information, I found strange the absence of data on topography/site (cardia vs noncardia).

33. Methods. Comparing overall survival between age-specific strata might not be extremely informative. Older patients have higher overall mortality than younger patients. The differences found are not surprising. A more objective comparison should be done with relative or, better, with net survival.

44. Discussion. In connection with major point no. 2. As age/histology differences were reported in the manuscript, I would expect a discussion on how H. pylori prevalence changes occurred in Germany and their role on interpreting the results also in combination on tumor sites. H.pylori infection could be attributable to 62% of tumors arising in the upper part of the stomach (cardia), and is a necessary cause for cancer development in the lower part of the organ (noncardia)(PMID: 34838195).

55. Results. Figure 1, I think it would be clearer to show boxplots or violin plots (please find here the open-source information for producing boxplot in R: https://r4ds.had.co.nz/exploratory-data-analysis.html?q=boxplot%20#cat-cont).

66. Discussion (second paragraph; lines 263-267). The authors’ critic of the literature can also extend to the results provided to the authors in this manuscript. Clustering the patients using age-percentile is a database related outcome, which compromises the final generalizability of the results. Given the sample size of the dataset, it would be good to set age-defined categories in the method (i.e., <50, 50-60, 60-70, >70) and evaluate/compare histology and tumor stage in those categories. This would be a procedure more scientifically objective which would lead to generalizable results. For instance, from Figure 2, we will find histology differences if we compare the oldest 50% of the patients with the younger 50%, but that comparison can also not be directly extrapolate/tested to other datasets (as it uses data empiric percentiles).

77.  Discussion. In connection with major points no. 3 and 6. The overall survival outcomes do not constitute a meaningful support for the clustering.

88. Discussion (lines 367-373). A lack of several important information is mentioned. What is the level of completeness of the information provided?

Minor:

11. Introduction (line 61). It is not clear that the incidence and mortality outcome are per year.

22. Lines 33 and 67. Two points: i) it should be early-onset gastric cancer, as not all the gastric cancer are carcinomas; and ii) please state the abbreviation completely so early-onset gastric cancer (EOGC).

33. Figure 2. Please consider a bar chart (so, you can also display absolute number for each proportion).

Author Response

Major:

Methods. The clustering of the patients is not described in the methods.

We thank the reviewer for this valuable observation. In fact, we missed to thoroughly describe the method of stratification in the “Methods” section as we reported the workflow in the results. We adjusted the manuscript accordingly and included a respective section in “Methods”. In addition, the term “combined approach of clustered age – histology comparison” might be somewhat misleading for the reader. Hence, we adjusted / clarified the approach taken in this manuscript.

Methods. Given the available information, I found strange the absence of data on topography/site (cardia vs noncardia).

We apologize for the lack of accuracy in our description of the methods / patient selection. Given several clinically and oncologically relevant differences between tumors at the cardia / gastroesophageal junction and tumors at other sites in the stomach (such as differing histology / histological subtypes, differences in gender and age distribution or differences in treatment algorithms amongst others), we decided to exclude specifically tumors at the cardia / gastroesophageal junction (ICD 16.0) from our analysis and to focus on gastric cancers in the remaining stomach. We clarified this in more detail.

Methods. Comparing overall survival between age-specific strata might not be extremely informative. Older patients have higher overall mortality than younger patients. The differences found are not surprising. A more objective comparison should be done with relative or, better, with net survival.

We thank the reviewer for this important comment. We highly agree that relative survival (or disease-free survival DFS) would provide additional important information on the population and the outcome. Unfortunately, DFS is included in the data set but completeness and plausibility are limited. We discussed this aspect in the ADT and decided that this information is not valid enough to be included into the manuscript. We added a respective comment to the “limitations of the study” section.

Discussion. In connection with major point no. 2. As age/histology differences were reported in the manuscript, I would expect a discussion on how H. pylori prevalence changes occurred in Germany and their role on interpreting the results also in combination on tumor sites. H.pylori infection could be attributable to 62% of tumors arising in the upper part of the stomach (cardia), and is a necessary cause for cancer development in the lower part of the organ (noncardia)(PMID: 34838195).

We thank the reviewer for this highly relevant and interesting point. Regarding tumors at the cardia please see above. With regards to H. Pylori: we fully agree that this information and analysis would strengthen the clinical relevance of the manuscript. Unfortunately, the data set provided by the German Clinical Cancer Registry Group does not include any information on H. Pylori status. For this reason, we did not include a section in the discussion on this topic as our paper does not provide any additional information in this regard. However, we mentioned this limitation in the discussion and can offer to add a section to the discussion highlighting the role of H. Pylori for cancer development in the stomach if requested by the reviewer.

Results. Figure 1, I think it would be clearer to show boxplots or violin plots (please find here the open-source information for producing boxplot in R: https://r4ds.had.co.nz/exploratory-data-analysis.html?q=boxplot%20#cat-cont).

We thank the reviewer for this suggestion and discussed the idea with our statistician. With figure 1, we aimed to provide a convincing simple “visual demonstration” that there is no “obvious peak” of incidence in the younger patients (both in the entire patient group as well as in the histological subtypes). Violin or boxplots would in our opinion not allow such an “easy visualization” of this “peak”. For this reason, we chose not to change the figure as suggested.

Discussion (second paragraph; lines 263-267). The authors’ critic of the literature can also extend to the results provided to the authors in this manuscript. Clustering the patients using age-percentile is a database related outcome, which compromises the final generalizability of the results. Given the sample size of the dataset, it would be good to set age-defined categories in the method (i.e., <50, 50-60, 60-70, >70) and evaluate/compare histology and tumor stage in those categories. This would be a procedure more scientifically objective which would lead to generalizable results. For instance, from Figure 2, we will find histology differences if we compare the oldest 50% of the patients with the younger 50%, but that comparison can also not be directly extrapolate/tested to other datasets (as it uses data empiric percentiles).

We thank the reviewer for this valuable comment. In fact, we initially established different age groups as suggested by the reviewer. However, after intense discussion in our team on how to potentially identify a subset of young patients that build a unique and somewhat homogenous group and differs from older patients, we decided to put the scope of this study on the attempt to objectively investigate if there is a young group of patients that can be identified not only by age, but by stratification of histology according to age percentiles. This approach using percentiles rather than pre-defined age cut-off points (e.g. <50 years of age, <60 years of age) was specifically applied in order to avoid “random selection” of an age cut-off point. We outlined the details of this approach in the methods. We absolutely agree that this approach does not allow easy comparison of our data to other publications using age-defined categories. However, comparison between all these studies is difficult as cut-offs are randomly chosen in most of the manuscripts, and the EOGC group in this publication is different to other reports. We did not intend to generate a database that allows comparison of all published data so far what means that we have to “re-calculate / re-build ” several EOGC groups with differing age cut-offs. We added this argument in the limitations of the study.

Discussion. In connection with major points no. 3 and 6. The overall survival outcomes do not constitute a meaningful support for the clustering.

Please see our answers to the respective comments above.

Discussion (lines 367-373). A lack of several important information is mentioned. What is the level of completeness of the information provided?

The number of patients included in the different analyses is provided in the text and the tables / graphs (n=xx). The “lack of several important information” refers to variables that are mostly not included in the data set.

Minor:

Introduction (line 61). It is not clear that the incidence and mortality outcome are per year.

We adjusted the manuscript by adding “yearly” at the respective positions.

Lines 33 and 67. Two points: i) it should be early-onset gastric cancer, as not all the gastric cancer are carcinomas; and ii) please state the abbreviation completely so early-onset gastric cancer (EOGC). 

We adjusted the manuscript according to the suggestions of the reviewer.

Figure 2. Please consider a bar chart (so, you can also display absolute number for each proportion).

We thank the reviewer for this suggestion. We decided to keep this form of presentation as it allows in our opinion easier visual access to the main argument of increasing rates of SRCC in patients with decreasing age.

Reviewer 2 Report

In the present study the authors have used patient data to characterized the EO and LO Gastric cancer. They have proposed to use age-histology to distinguish and define Early onset patients and late onset patients. The rest of study presents data summary of features from these stratified groups (EO vs LO).  For making a broader appeal I would request authors to include few lines on how these insights can help in improvement of prognosis, diagnosis and/or treatment of EO and LO patients. 

I am also curious, once the working definition of EO is established, can the authors elaborate if there are other features or their combinations which result in similar EO-LO classification or sharpen the boundary of distinction.

Author Response

In the present study the authors have used patient data to characterized the EO and LO Gastric cancer. They have proposed to use age-histology to distinguish and define Early onset patients and late onset patients. The rest of study presents data summary of features from these stratified groups (EO vs LO).  For making a broader appeal I would request authors to include few lines on how these insights can help in improvement of prognosis, diagnosis and/or treatment of EO and LO patients.

We thank the reviewer for this valuable comment. We added a respective section to the conclusions highlighting the need for an increased awareness and early diagnostic workup as well as liberal indication for more aggressive treatment.

I am also curious, once the working definition of EO is established, can the authors elaborate if there are other features or their combinations which result in similar EO-LO classification or sharpen the boundary of distinction.

This aspect is highly interesting. Multivariate analysis demonstrated a number of factors such as tumor stage or grading to predict outcome in these patients. However, these factors are only available once the diagnosis is made, and most of these factors impact on each other (e.g. T-stage and nodal involvement) or depend on histological subtypes such as diffuse type or signet ring cell carcinoma limiting therefore in our opinion the usefulness for better classification. Yet, there are other very interesting candidates that might impact on better classification of these patients, including for example H. Pylori status, recent gastritis or similar. Unfortunately, the data set used for this analysis does not include data in this context.